

# How urban impervious surface shapes bird foraging behavior in an arid city

Simone Vega Rabelo[1], Jeffrey D. Haight[2] and Heather L. Bateman[1]

[1] College of Integrative Sciences and Arts, Arizona State University, Mesa, AZ, United States of America
[2] Global Institute of Sustainability and Innovation, Arizona State University, Tempe, AZ, United States of America

## ABSTRACT

Wildlife behavior and interactions in urban ecosystems can vary across landscape types and species, with some birds dominating human-derived resources. This study investigates the relationship between urbanization, measured as impervious surface cover, and bird foraging behavior in the Phoenix metropolitan area in Arizona, USA. We conducted 126 feeding trials across 13 sites along a gradient of urbanization and bird interactions with anthropogenic food sources present. Trials were conducted using popcorn placed at random distances and orientations from a trash can. We recorded bird species visiting feeding stations and time to first visit during 20-minute trials and then quantified relationships among visitation timing (latency), species richness, and impervious surface coverage. Time to first visit was negatively correlated with the amount of impervious surface, with the highly urban areas having birds arrive at the feeding station the soonest. Great-tailed Grackles (*Quiscalus mexicanus*) comprised the most common visitor across all impervious surface levels. Non-native doves like Rock Pigeons (*Columbia livia*) and Eurasian Collared-doves (*Streptopelia decaocto*) were quicker to visit feeding stations compared to native doves like Mourning Doves (*Zenaida macroura*), Inca doves (*Columbina inca*), and White-winged Doves (*Z. asiatica*). Small urban-adapted generalists, like House Sparrows (*Passer domesticus*) and House Finches (*Haemorhous mexicanus*), tended to be more frequent visitors at highly urbanized sites compared to larger birds. These emphasize how species-specific foraging behaviors can differentiate resource use by birds in urban areas.

## INTRODUCTION

As cities often overlap with areas of high species richness, wildlife habitat can be threatened by areas of rapid urban growth (*Cincotta, Wisnewski & Engelman, 2000*). However, urban green spaces can be important to support biodiversity by providing areas where wildlife and humans can coexist (*Aronson et al., 2017*). Metropolitan areas often contain a mix of native and non-native species, with species adapting to urban landscapes with varying levels of success (*Shochat et al., 2004*). Urban wildlife communities can be shaped by the availability of food and shelter, among other ecological factors (*Callaghan et al., 2019a*). Although some species struggle to adapt to urban conditions, birds may thrive because of traits associated with novel landscapes and anthropogenic resources (*Evans et al., 2011*).

Corresponding author
Heather L. Bateman,
Heather.L.Bateman@asu.edu

For avifauna, habitat alterations that produce anthropogenic waste that can attract certain bird species while deterring others (*García-Arroyo, Gómez-Martínez & MacGregor-Fors, 2023*). Birds in urban environments often encounter abundant food sources, including food waste left behind by humans (*Brown et al., 2022*). This advantage allows "urban exploiters" to outcompete "urban avoiders" in these environments (*Blair, 1996*). As a result, urbanization can alter the composition and behaviors of local bird populations (*Pena et al., 2023*).

Behavioral responses to novel stimuli can determine how birds exploit anthropogenic food sources in urban environments. Animals can display neophilia, which is the attraction to a novel stimulus, and neophobia, the avoidance of a novel stimulus (*Tryjanowski et al., 2016*). In cities, where food items are frequently unfamiliar or irregular, neophilic species scavenge human food waste, a trait among common and widespread urban-adapted birds (*Brown et al., 2022*; *Pejchar et al., 2025*). The heterogeneity of urban settings can desensitize birds to perceived foraging risks, encouraging bolder behavior (*Tryjanowski et al., 2016*). For more neophobic birds, individuals will quit foraging when the energetic benefits no longer outweigh the costs (*Shochat et al., 2004*). Consequently, smaller generalist species such as House Sparrows (*Passer domesticus*) often capitalize on feeding opportunities more quickly than larger species, which tend to assess risks differently and are more easily displaced by anthropogenic activities (*Haemig, De Luna & Blank, 2021*). Together, these patterns highlight the ecological strategies birds employ to navigate the risk and rewards of the urban landscape.

Urban landscape characteristics, such as the density of buildings, pavements, and impervious surfaces, play a crucial role in shaping the presence and behaviors of wildlife species. These features can alter the availability of essential resources like food, water, and shelter (*Callaghan et al., 2019a*) and even influence the presence of wildlife diseases (*Hernandez et al., 2016*). Specifically, impervious surfaces are key indicators of urbanization-driven habitat loss associated with reduced presence, abundance, and diversity of wildlife taxa, especially among bird species (*Aronson et al., 2017*; *Haight et al., 2025*). Urbanization is also characterized by a reduction in vegetation cover, as measured by NDVI (normalized difference vegetation index) which is inversely related to the amount of impervious surface area (*Weng & Lu, 2008*). Therefore, impervious surfaces could affect the foraging behaviors of birds in urban environments by altering resources and vegetation density, which may influence the types of birds that visit these areas.

In this study, we conducted a field experiment to investigate how bird foraging behavior can vary across an urban gradient, defined by proportion of impervious surface. We specifically assessed: time (latency) for birds to visit a feeding station across a gradient of impervious surface. We then considered species-specific likelihood of visits to feeding stations across impervious surface levels. In general, we predicted that non-native birds would visit stations sooner and at locations with higher levels of impervious surface. We hypothesized that smaller species, such as the House Sparrow, would visit feeding stations sooner in high-impervious areas, while larger species and species associated with expanding their range into mesic habitats like the Great-tailed Grackle (*Quiscalus mexicanus*), would visit feeding stations sooner in areas with lower impervious surface.
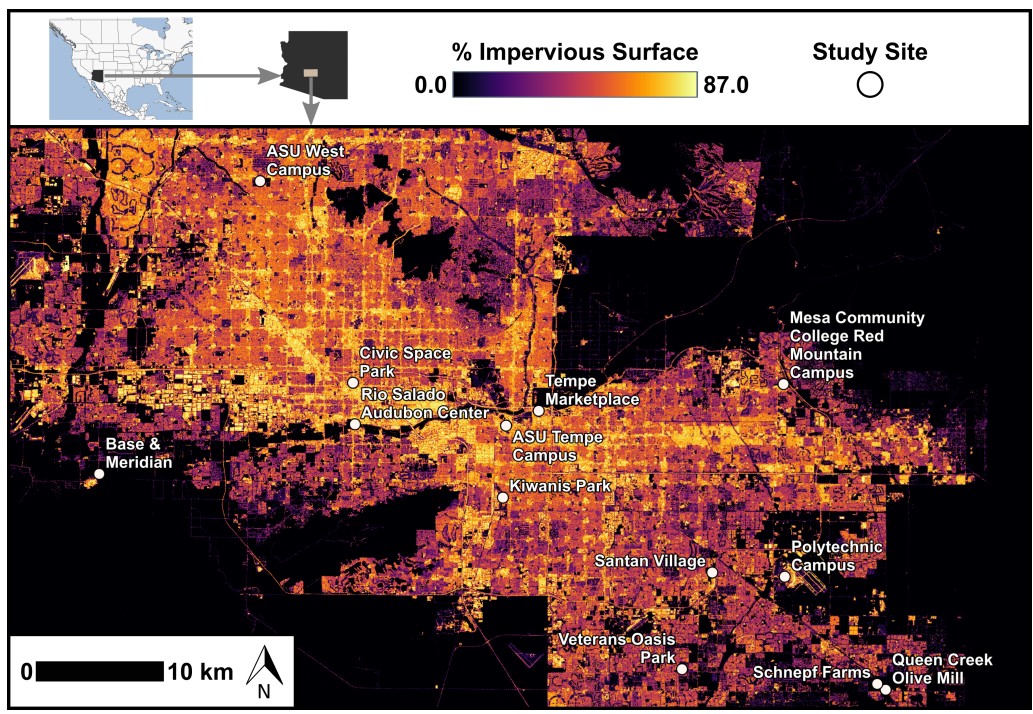

**Figure 1** **Map of feeding trial locations across range of impervious surfaces.** Locations of 13 sites with feeding trials within the Phoenix metropolitan area, Arizona, USA. Values of impervious surface are depicted here based on the 2021 National Land Cover Database (*United States Geological Survey, 2024*). We characterized impervious surface in our analyses using the i-Tree Canopy tool (*i-Tree, 2024*). Study site locations ranged in longitude from −112.306 to −111.583 and in latitude from 33.221 to 33.608.

## MATERIALS & METHODS

### Study area

This study was conducted in the Phoenix metropolitan area (metro Phoenix; Fig. 1), characterized by its location within the Sonoran Desert ecosystem (*Comus et al., 2015*). This desert environment is known for its hot and arid climate, sparse natural shrubland vegetation, and landscapes transformed by urbanization, including the creation of impervious surfaces such as buildings and roads. The urbanized and agricultural environments of metro Phoenix are also characterized by extensive year-round irrigation practices that support greater vegetation cover and productivity than that of natural desert habitats (*Buyantuyev & Wu, 2009*).

### Feeding trials

We assessed bird foraging behavior by establishing feeding stations at 13 sites across a gradient of urbanization. Feeding stations were placed randomly within 13 sites across the metropolitan Phoenix area and were selected for public accessibility and with a range of (24% to 90%) urbanization levels (Fig. 1). Sites included the Rio Salado Audubon Center in Phoenix, Kiwanis Park in Tempe, the Queen Creek Olive Mill, Veterans Oasis Park in Chandler, Base and Meridian Wildlife Area in Avondale, Schnepf Farms in Queen Creek,
the ASU Polytechnic Campus in Mesa, the Mesa Community College Red Mountain Campus, the ASU West Valley Campus in Glendale, Civic Space Park in Phoenix, the ASU Tempe Campus, Santan Village Mall in Gilbert, and Tempe Marketplace. Levels of impervious surface at each site were calculated post-selection using i-Tree Canopy (*i-Tree, 2024*), an online tool that provides estimates based on satellite imagery (explanatory variable described in landscape variables).

Time to first visit was recorded during 6-16 trials per site to observe bird behavior near anthropogenic waste sources (a total of 126 feeding trials). Trash cans were the primary starting points for positioning feeding stations because they are ubiquitous in the urban area, represent places of resources for urban wildlife (*García-Arroyo, Gómez-Martínez & MacGregor-Fors, 2023*), and to maximize geographical coverage at each site. We did not include large dumpsters in this study. In areas with limited or no trash cans, alternative human-made structures (*e.g.*, benches or signs) or visible trash on the ground were used as starting points, these made up fewer than 8% of trials. Most sites had 10 feeding stations. Smaller sites had fewer than 10 stations (*i.e.,* Mesa Community College Red Mountain Campus had six trials, Civic Space Park had seven trials, and Base and Meridian Wildlife Area had seven trials), and one large site had more than 10 stations (*i.e.,* the ASU Tempe Campus had 16 trials).

Once a starting point was chosen, a feeding station was established by adding a handful (15 pieces) of minimally processed and unseasoned popcorn placed on the ground. The observer walked away (approximately 25 m away) from the station and used binoculars for observations. We positioned feeding stations at a random distance (5–20 m, using a random number generator) with a random orientation from the starting point. Random orientations were chosen by dividing the area around the trash can into 12 directional segments (*e.g.*, a clock with N at 12:00) and used a random number generator to select one. For the safety of observers, we avoided establishing a feeding station in a parking lot and therefore truncated the orientation to avoid these areas.

Observations lasted for 20 min during early morning hours (6:00–9:30 AM, although some surveys exceed this range) to maximize bird activity in a hot, arid environment. We recorded bird species visiting feeding stations through visual identification (based on *Sibley, 2022*) during the 20-minute trial periods. All birds landing at or approaching the feeding stations were noted. For stations with no visitors, a maximum time of 20 min (1200 s) was recorded as the first visit time. For the first month of surveys (12%, or 15 of 126), we only recorded data for the first bird visitor to each station, coding the visitation of that first species as '1' and all other species to that station as 'NA'. If no visitors were observed at a station, then visitation for all species was recorded as '0'. After the first month, we also recorded birds that arrived after the first visit within the entire 20-minute survey period (88%, or 111 of 126 stations), noting whether or not each species visited the station at any point (coded as 1 or 0) and calculating species richness as the sum of all visiting species. We excluded 'NA' values from analyses of species richness and of species-specific station visitation described below. Field data collection occurred over a 14-week period, from 7 July 2024 to 13 October 2024. This period overlapped with bird migrations, such as the White-winged Doves (*Zenaida asiatica*) migrating from Arizona to Mexico in the

fall. However, most bird visitors were non-migratory, resident species. ASU Institutional Animal Care and Use committee approved methods for this research (23-2016T).

### Landscape variables

We related bird foraging behavior, species richness, and likelihood that individual species would visit a station to levels of urbanization by quantifying percent impervious surface. We quantified landscape characteristics at each site using the *i-Tree (2024)*, which estimates impervious surface cover and vegetation features based on satellite imagery. The tool produced by the US Department of Agriculture utilizes imagery datasets from Google Earth. First, a bounding box was drawn around the sampling areas for each site. Then, 50 random land cover sampling points were placed across the sampling area (manually drawn grid and bounded site by roads), and each point was manually characterized by its land surface cover (*e.g.*, impervious road, impervious buildings, tree/shrub, grass/herbaceous vegetation). We combined the resulting percentages of each land surface cover type to define impervious surface as including roads, buildings, and other impermeable land cover types (*e.g.*, concrete sidewalks). For analysis purposes, all feeding stations at each site were assigned the single impervious surface value for that sampling area.

### Data analysis

We investigated research questions by modeling the likelihood that individual species would visit a feeding station, the time to first visit, and species richness in relation to impervious surface levels. Feeding station-level response variables included whether or not a species or group of species (*i.e.,* native or non-native doves) visited a station, latency or time to first visitation, and species richness (total number of species that visited a station). We quantified relationships of each response variable with impervious surface area, day-of-year (DOY; Julian date of feeding trials at each site) using generalized linear mixed models (GLMMs) fit within the R programming language 4.1 using the *glmmTMB* package (*Brooks et al., 2017*; *R Core Team, 2024*).  We modeled the responses of whether or not a species or group of species visited, latency, and species richness as having binomial (logit-link), normal (Gaussian-link), and Poisson (log-link) distributions, respectively. We fit GLMMs for each response variable with two sets of fixed effects (impervious surface area only, and impervious surface area and DOY) and study site as a random effect (intercept). We standardized each fixed effect covariates prior to model fitting and present standardized model coefficients ($\beta$). We compared the relative quality of models with and without the DOY fixed effect by using the R package *performance* to calculate marginal $R^2$ values for fixed effects and the Akaike Information Criterion adjusted for low sample size (AICc), regarding the lower-AICc model as being the better model (*Anderson & Burnham, 2002*). Impervious surface area and DOY did not demonstrate significant collinearity across 13 sites ($r = 0.115$, $p = 0.709$).

## RESULTS

### Bird visitors

Across all trials, a total of 15 bird species were observed visiting feeding stations (Table S1). Out of 126 trials, 64 trials had no visitors (50.8%) and 62 (49.2%) had visitors. Of the
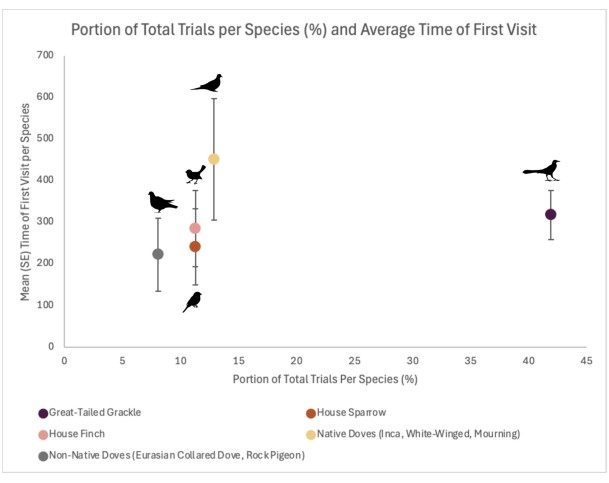
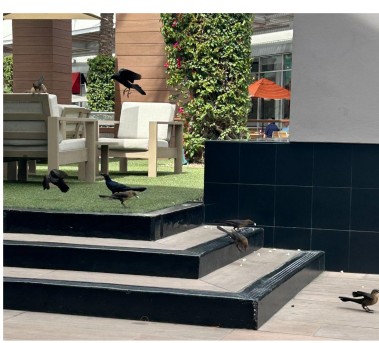

**Figure 2** **Five most common species/groups seen during feeding trials at stations with visitors across 13 sites in the Phoenix metropolitan area, Arizona, USA.** Mean in seconds (and SE) to first visitor to the feeding station (popcorn). Non-native doves include Eurasian Collared-dove (*Streptopelia decaocto*) and Rock Pigeon (*Columbia livia*). Native doves include Mourning Doves (*Zenaida macroura*), Inca Dove (*Columbina inca*), and White-winged Dove (*Z. asiatica*). The most common visitor was Great-tailed Grackles (*Quiscalus mexicanus*; photo).

trials with visitors, the first visits were recorded as follows: 41.9% were Great-tailed Grackles, 12.9% were native doves (Inca Dove, *Columbia inca*; White-winged Dove; and Mourning Dove), 11.3% were House Sparrows, 11.3% were House Finches (*Haemorhous mexicanus*), 8.1% were non-native doves (Eurasian Collared-dove, *Streptopelia decaocto*, and Rock Pigeon, *Columbia livia*), and 3.2% were Abert's Towhees (*Pipilo aberti*). The following species were recorded only once as the first visitor: Rock Wren (*Salpinctes obsoletus*), Cactus Wren (*Campylorhynchus brunneicapillus*), Northern Mockingbird (*Mimus polyglottos*), Greater Roadrunner (*Geococcyx californianus*), Curve-billed Thrasher (*Toxostoma curvirostre*), and European Starling (*Sturnus vulgaris*). Most trials had only a single species visit the feeding station. For example, in trials where all species were counted (111 trials), the mean richness was 1.3 birds and one was the median value.

Visitation behaviors at feeding stations varied among bird species. Great-tailed Grackles were the largest proportion of first birds to visit feeding stations and averaged 318.0 s (58.45 SE) to arrive (Fig. 2). Among the common groups of birds that visited, native doves took the longest to arrive and averaged 451.3 s (145.98 SE), then House Finches averaged 285.7 s (91.50 SE), and House Sparrows averaged 242.0 s (91.53 SE). Non-native doves were the quickest to visit feeding stations and averaged 222.6 s (87.86 SE; Fig. 2).

### Bird behavior related to impervious surface

Bird visitation to feeding stations occurred across a gradient of 24% to 90% impervious surface area. Across all response variables, the best model was the one including the fixed effect of impervious surface area ($\beta_{ISA}$) but excluded DOY (Table S2). Three species or groups of species (native and non-native doves) differed in their likelihood to visit stations across levels of impervious surface. Native doves were more likely to visit less urbanized

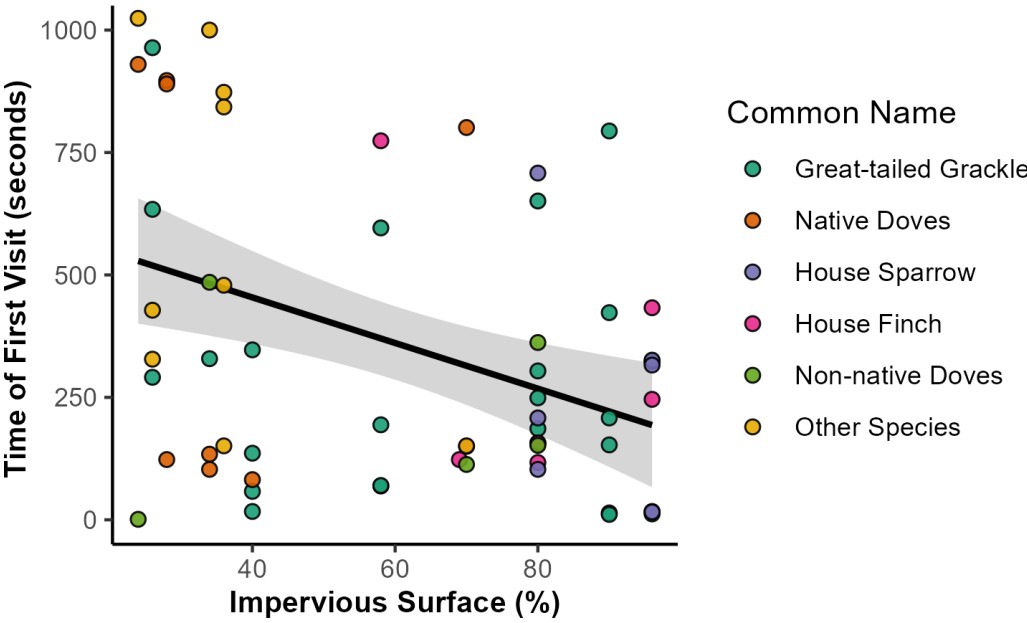

**Figure 3  Relationships of impervious surface area with time of first bird to feeding station.** Relationship of impervious surface area with time of first bird feeding station visits. Out of 126 feeding stations, a total of 62 were visited by any bird species. Across all species, time of first visits (in seconds) was negatively associated with sites with higher percentage of impervious surface. Point colors indicate the species identity of the first visitor to each station (in order of visitation frequency), with some species grouped for visualization purposes: native doves (Mourning Dove, White-winged Dove, Inca Dove), non-native doves (Eurasian Collared-dove, Rock Pigeon), and other species (Abert's Towhee, Rock Wren, Cactus Wren, Northern Mockingbird, Greater Roadrunner, Curve-billed Thrasher, and European Starling). Shaded area represents the 95% confidence interval of the trendline, as approximated by the 'geom_smooth' function in the R package 'ggplot2'.

feeding stations ($p = 0.042$, $\beta_{\text{ISA}} = -1.18$, 95% CI [$-2.31, 0.04$]). More urbanized feeding stations were more likely to be visited by House Finches ($p = 0.047$, $\beta_{\text{ISA}} = 1.06$, 95% CI [$0.02, 2.10$]) and by House Sparrows ($p = 0.029$, $\beta_{\text{ISA}} = 0.148$, 95% CI [$0.15, 2.81$]). Impervious surface area had no significant relationship with initiation by Great-tailed Grackles ($p = 0.277$, $\beta_{\text{ISA}} = 0.28$, 95% CI [$-0.23, 0.79$]) or non-native doves ($p = 0.790$, $\beta_{\text{ISA}} = -0.10$, 95% CI [$-0.81, 0.62$]). The best model for latency indicated that birds visited feeding stations faster in areas with high levels of impervious surface ($p = 0.001$, $\beta_{\text{ISA}} = -118.52$, 95% CI [$-188.78, -48.26$]; $R^2 = 0.152$), with the average time for the first visitor to arrive at feeding stations with the highest impervious surface being 2.7 times shorter than the areas with lowest impervious surface (Fig. 3). Latency time demonstrated no substantial relationship with DOY, as indicated by the relatively low quality of models containing the DOY fixed effect (Fig. S1; Table S2). We detected no significant relationship between species richness and impervious surface ($p = 0.778$, $\beta_{\text{ISA}} = 0.034$, 95% CI [$-0.20, 0.27$]).

## DISCUSSION

We explored urban bird foraging behavior by conducting a field experiment establishing feeding stations across a range of impervious surface levels in the greater Phoenix metropolitan area. About half of the stations had bird visitors and richness was low with only a single species visiting during most trials. In places with high amounts of impervious surface, birds visit feeding stations sooner compared to places with less impervious surface, with Great-tailed Grackles being the most common first visitor (over 40%) and likely the species contributing most strongly to the overall trend in latency (Fig. 3). Other species, such as native doves, took longer to arrive, whereas non-native doves were the quickest to visit. Bird species also varied in foraging behavior across the urban landscape. The likelihood of native doves visiting stations had a negative relationship with impervious surface amount; whereas, House Finches and House Sparrows had the opposite pattern and were more likely to visit stations with high levels of impervious surface.

Urbanized environments can be advantageous for birds with generalist feeding behaviors, especially species that can use anthropogenic food sources (*Seress & Liker, 2015*). In our study, feeding stations in more urbanized areas were visited quickly by urban-adapted, generalist species (*e.g.*, Great-tailed Grackle, non-native doves, House Sparrows). These results are consistent with others showing that urbanization tends to select bird species with more flexible, generalist diets (*Hahs et al., 2023*). Great-tailed Grackles are considered native to North America and are a species that have thrived in landscapes converted to agriculture and irrigated vegetation in dryland ecosystems. The species has expanded its range in North America since the late 19th century due to human-driven landscape changes by thriving in urban areas by exploiting anthropogenic food sources like waste grain and discarded food (*Wehtje, 2003*). Grackle presence in the Phoenix metropolitan area is linked to cultivated lands and wetland environments within urban areas where they have benefited from reduced nest predation and stable food supplies (*Wehtje, 2003*). Grackles' ability to exploit novel food sources, including livestock feedlots, allows them to thrive in urban environments, making them well-adapted to human-altered landscapes in arid lands (*Pandolfino & Handel, 2018*).

During our trials, we observed that first visitors often monopolized the feeding station, preventing access to other birds. Although this behavior was not limited to a single species, it was commonly observed from Great-tailed Grackles, similar to other feeding studies (*Fronimos et al., 2011*). The tendency of first visitors to dominate the feeding station and prevent access to others may help explain the lack of differences in species richness across sites. In our study, all visitors were birds. Perhaps by expanding sampling beyond urban areas to rural lands, other non-avian taxa, such as small mammals, could be included such as in the study by *Swartz, Blaney & Behm (2024)*. Future studies could investigate how dominance hierarchies or non-avian species structure urban bird communities.

As urbanization continues to alter resource type and availability, species with high behavioral flexibility and generalist feeding strategies are likely to dominate, potentially outcompeting more specialized native species over time (*Carlon & Dominoni, 2024*). Non-native doves like Rock Pigeons and Eurasian Collared-doves were associated with

more urbanized areas in this study compared to native doves like Mourning Doves, Inca doves, and White-winged Doves. Generalist species, like non-native doves, House Sparrows, and House Finches are more urban-tolerant due to their broader ecological niches and behavioral flexibility (*Callaghan et al., 2019b*). Urban-exploiting species have been associated with areas having high density of restaurants and discarded food waste (*Brown et al., 2022*). Research has shown that urban birds generally display higher neophilic tendencies than rural birds, enabling them to take advantage of unpredictable resources in urban areas (*Tryjanowski et al., 2016*). Thus, urbanization may shape avian community composition by altering how communities are formed, leading to a loss of specialists and less diverse ecosystems (*Carlon & Dominoni, 2024*).

Sonoran Desert birds, such as Cactus Wrens and Curve-billed Thrashers and winter migratory species such as White-crowned Sparrows (*Zonotrichia leucophrys*), were rarely observed at feeding stations in this study, suggesting that urbanization may exclude some species from anthropogenic food resources. One way the methods in this study could have excluded species was the choice of food at feeding stations. Some insectivorous species may not have viewed popcorn as a food source. Future studies could use a different food to determine how insectivorous birds relate to novel food sources in urban areas.

The relationship between urbanization and bird foraging behavior has implications for balancing the coexistence of humans and wildlife in cities. Birds play a wide range of social-ecological roles in urban ecosystems, and their traits and behaviors can influence human well-being. The species that are more likely to scavenge food waste in urbanized settings (*e.g.*, Great-tailed Grackles, House Sparrows) are commonly characterized as "messy" birds with unpleasant appearance or sound (*Brown et al., 2022*), traits associated with negative attitudes toward and diminished appreciation of their ecosystem services (*Andrade et al., 2022*).

## CONCLUSION

Overall, this field-based investigation found that urban birds utilize anthropogenic food resources quickest in places with high impervious surfaces. The species composition includes both native and non-native species and the most common species, Great-tailed Grackles, are a common urban adaptor in arid land systems. Although many studies assess the value of wildlife (*Von Döhren & Haase, 2015*), understanding the relationship between avian ecology and human perceptions can better manage urban ecosystems. For humans and wildlife to coexist in urban systems, it is essential to consider the nuanced positive and negative impacts of species traits and biodiversity on human well-being. By acknowledging these complexities, we can promote coexistence and tolerance of urban wildlife.

## ACKNOWLEDGEMENTS

Karen Sweazea provided suggestions on bird feeding methods. We thank two peer reviewers for providing feedback to improve the manuscript.

### Funding

This research was supported by the National Science Foundation through the Central Arizona-Phoenix Long-Term Ecological Research Program (grant no. DEB-2224662). There was no additional external funding received for this study. The funders had no role in study design, data collection and analysis, decision to publish, or preparation of the manuscript.

### Grant Disclosures

The following grant information was disclosed by the authors:
Central Arizona-Phoenix Long-Term Ecological Research Program: DEB-2224662.

### Competing Interests

The authors declare there are no competing interests.

### Author Contributions

- Simone Vega Rabelo conceived and designed the experiments, performed the experiments, analyzed the data, prepared figures and/or tables, authored or reviewed drafts of the article, and approved the final draft.
- Jeffrey D. Haight analyzed the data, prepared figures and/or tables, authored or reviewed drafts of the article, and approved the final draft.
- Heather L. Bateman conceived and designed the experiments, prepared figures and/or tables, authored or reviewed drafts of the article, and approved the final draft.

### Animal Ethics

The following information was supplied relating to ethical approvals (i.e., approving body and any reference numbers):

ASU Institutional Animal Care and Use committee approved methods for this research (23-2016T).

### Field Study Permissions

The following information was supplied relating to field study approvals (i.e., approving body and any reference numbers):

Schnepf Farms and Queen Creek Olive Mill allowed access. Access permission for Schnepf Farms was verbally given by an anonymous Schnepf Farms employee on the day of the site visit. Permission for Queen Creek Olive Mill was given over email (please see attached confidential Supplemental File).

### Data Availability

The experimental data is available in the Supplemental File.

### Supplemental Information

Supplemental information for this article can be found online at http://dx.doi.org/10.7717/peerj.19980#supplemental-information.

# PeerJ

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
