# Peer review of "How urban impervious surface shapes bird foraging behavior in an arid city"

_PeerJ, doi:10.7717/peerj.19980_

## Round 0.1 · original submission · Major Revisions

I am happy to report we have received two positive reviews, and the Reviewers have provided some excellent recommendations for improving this manuscript. I think it is important that the authors, while revising the manuscript pay particular attention to addressing each point made and genuinely taking the suggestions and comment to heart.

Notably, some of the areas that would benefit the most during the next round of revisions relate to writing style and format, paying close attention to how the study design is presented and reported. I recommend more senior authors and supervisors apply the experience here to help advance these aspects. Overall, this is an interesting study, and it certainly has the potential to be a solid article.

In my reading of the paper, and from what I see based on Reviewer 1’s and 2’s comments, I think there are two major themes, refocusing and rejigging both the introductions and discussion and ensuring that they are better addressing their questions. In general, some aspects read as too vague, while others make claims that the data or approaches cannot support (i.e., reaching too far).

Address the manuscripts ‘bookends’ (i.e., the Introductions and Discussion) significant attention needs to be paid to providing more context and content. Applying a more formulaic approach to scientific writing would enhance theses areas. The goal of the Introduction is to set up all of the proverbial breadcrumbs so that the reader has all the required information so that once the findings are revealed in the Results, that they understand the idea put forward in the discussion. So here, be thorough and clear in setting your breadcrumbs. Each introduction paragraph needs a strong topic sentence, followed by a series of assertions, examples (and even contradictions) to the ideas covered within the paragraph’s topic. Then conclude the paragraph with a summary of the topic covered and finish with a segue to the next topic that will be presented in the following paragraph. As such Intro/Discussion paragraphs are robust, yet still to the point (no fluff or tangents) – the goal is to set a topic covering it in a comprehensive way that sets up the reader for a better deeper dive into the topic that will be presented next. As such, paragraph fragments such as those with 4-6 sentences are not complete enough. Not in content, or context. The same goes for the discussion. I strongly recommend you overhaul these bookends.

A good starting framework for the Intro would be as such:
P1: General topic
P2: Subsection of general topic of special interest, why is it interesting
P3: What are some knowledge gaps of the interesting topic subsection
P4: What is the model system you have that would allow you to test these knowledge gaps
P5: How are you going to address a knowledge gap, your hypothesis, predictions and/or questions

A good starting framework for the Discussion would be as such:
P1: Summarize your results and relate them to your predictions/hypotheses
P2-4: Subsection for each theme of your paper (based on the study goals and dictated by your predictions/hypotheses). Make sure to put them in context with literature (where is fits and where it does not). Point to what need to be examined next given what you found and know
P5: How does this work fit into the larger field of study for urban evolutionary ecology, where ideas of theories does it advance, and what sorts of new questions does these finds give rise to.
P6: Conclusion

Similarly, we need to make sure that the goals of this study ae not only clearly stated at the end of the introduction (which they are) – but that this clarity is seen across the manuscript (i.e., in the Data Analysis section in Methods, Results, and each of the three ideas is the clear topics of their own body paragraph in the discussion). Subheading for each of the three topics in each of these listed section may be a helpful approach to stay on target. Be clear, provide robust content to provide enough context, and ensure you are always connecting to the purpose of the study (i.e., the hypothesis/predictions/goal/etc.). The reviewers have some excellent suggestions here and I strongly recommend you take them on.

Overall, I think this study is fascinating that has potential to be formed into a solid article. What it needs now is some much greater attention be paid to how this work is being presented. Writing structure, formatting, and content needs to be addressed. These are skills that come with practices and experience, and we urge those authors on this paper with that experience to guide those learning the proverbial ropes here, so that this draft manuscript can be improved and polished to a more professional version of this manuscript. Please ensure this happens so that we can continue working with you to provide feedback so that this study can be continually polished into the shine paper it deserves to be.

·

Basic reporting

Clear mostly unambiguous English, sources are cited and conforms to standards of professional expression and article structure.
I have made some specific comments on the manuscript below for writing and clarity issues, but I found that the intro and discussion were too long for the complexity of the results presented, and not clear to me exactly what the driving questions were. Consider more careful concise language, but additional points will need to be explored when addressing the experimental design component of the manuscript. Moreover, the discussion the writing is not focussed enough on how the results fit with their hypotheses stated in the last paragraph of the introduction, and the authors do not effectively incorporate their findings with the literature, and mostly just reviews the literature on birds featured in the data.
Overall, the methods and results sections need to more clearly link how these data analytical approaches actually answer their stated research questions. This can be done with the use of careful topic sentences and reminding the reader of central ideas.
The results are mostly relevant to the some of the stated hypotheses, but there are some flaws that I will discuss in the next section that must be addressed before publication.

Experimental design

The research is original, and with some reframing and some different analytical approaches this could fit within the scope of the journal.
The authors are broadly interested in how bird foraging behaviour is linked to urban bird community composition and ecosystem services. The authors use % impervious surface as a metric for degree of urbanization and suggest they are interested in the ecosystem service of trash removal. They did this by setting up a series of replicated urban trash removal trials across Phoenix, well done on completing that difficult work!
The authors then state they examine the relationship between:
1) time to first visitation across % impervious surface
2) species composition of first visitors and % impervious surface and
3) bird body size and time to first visitation
However, the authors did not effectively address these questions which I will discuss in more detail below. Moreover, while these are laudable questions in of themselves relating to bird community and behaviour, the experimental design does not actually measure the service of trash removal which they say is their overarching aim. To address the ecosystem service question the authors would need to have measured the duration of birds at the station, and/or how much birds consumed, as well as measure the total amount of popcorn consumed at the end of the trial.
Broadly, the methods are stated in such a way that they are replicable, but I had specific comments about parts of the methods writing which would influence replicability (If it’s unclear then I can’t replicate it). Moreover, the “bird community composition” section feels redundant because they do not further analyse the data on composition aside from species of first visit.
Also, I think it is important know if there an effect on the type of trial starting point on the results. Seems to me that trash cans would be a permanent source of food, but these other locations would be less likely a “known food source”. Thus these two starting points could have different arrival times just based on whether birds recognize a location as a reliable food source.

Validity of the findings

Broadly, I think the methods are not appropriate for answering the stated questions.
The authors collected time to first visit across multiple replicated trials at a location, and then took the average of the time to first visit. I believe they then used these mean per site data as the inputs for their glms. I do not think that this is appropriate and it loses a lot of the richness of the data. I would consider exploring mixed effects models to enable the full dataset to be explored here.
Further the authors explore the time to first visit per species as well as the frequency of first visitation in figure 2, but do not make any attempt to determine whether there is a statistical difference between species. Moreover, to better address their questions the authors would do well to try to disentangle whether the dominant species is more important for time to visitation, rather than impervious surface. Because there is nothing in the methods and results that actually links species composition to % impervious surface, despite it being a stated question in the introduction (2). There are a number of ways to approach this, such as including ecological diversity indices, and/or exploring how % impervious surface affects visitation time within each species.
I don’t really think that their approach to exploring how bird body size relates to visitation time is appropriate, given that there is no attempt to control for how frequent a species occurs at a site. If grackles are dominant first visitors this is going to be reflected in the mean body size for the site. Moreover, the authors state that they found a relationship between body size and visitation when this result was non-significant.
Further, there are data reporting issues within this manuscript which mean that the technical standard is not quite met. Specifically the raw data were inadequately presented and does not easily enable replication. The raw data includes two spreadsheet pages, which are not clearly labelled. The second tab is just called “Jeffs data”, and is not an appropriate description. After some fiddling, I realized that “jeffs data” could just be presented in the same data sheet as the main data. Moreover, column titles are inadequate or incorrect (e.g Site Number should be trial number), and authors report the mass of the first visitor in lbs, yet the manuscript uses grams (hint, it should all be reported in grams). These are fairly easy to fix.

Additional comments

Specific comments:
Line 81: Not the correct citation for optimal foraging theory
99: technically this paper does not measure visitation rate, but time to first visitation (or latency). These are not the same thing.
123-126: At what kind of spatial scale is the % impervious surface calculated at? 1km2?
The discussion with mentors explanation is not sufficient for determining adequate replication and variation. The figure is nice, but address the variation and replication issue, it would be good to discuss how many sites in a % impervious surface bin (such as high medium low) was considered adequate replication. Or, at minimum, what was the minimum and maximum % impervious cover.
130-133: The choice of small trash cans to maximise geographic coverage doesn’t make sense to me? What does this mean? It seems to me that the sites were selected on the availability of existing trash and thus a pre-existing signal of food availability??

135-138: This is confusing. You have so many more sites than mentioned in this list. Was there a certain number of trials that you were aiming for and the majority were at that number of trials?
164-171: Seems like this should go in the study area section. As per comment on Ln 123-126, this still doesn’t give the spatial extent of the site. If this varied 50 random points within in a large vs small radius could represent different things.
174-175: It is more helpful to remind the reader of the research questions to explain why you did a particular analysis.
176-177: do not need to describe the formula for average.
185: Need to include what distributional assumptions you included in the model? Was it a gaussian model? Did you use a particular R-package?
230-231, flip the clauses in this sentence ie. We explored x by running experiment…
First paragraph of discussion: This paragraph should mostly re-state how your results relate to your main hypotheses, but it does not do this effectively.
267: socio-ecological not social-ecological.

Final paragraph of the discussion: It may be true that people view grackles as messy or unpleasant (but why they’re so good?!), but they ARE native. The language in this section needs to be more precise to reflect this, because at the moment it suggests that grackles are non-native.

290-291: The conclusion that: “Urban planning that increases green spaces and more mesic landscapes may promote these undesirable species and support fewer native species, especially in arid land ecosystems”, is not supported by the data analysis. The authors did not convincingly demonstrate that a particular species was associated more heavily with a % impervious surface.

Reviewer 2 ·

Basic reporting

The manuscript is clearly written in professional and fluent English, with clear structure and appropriate formatting. Figures and tables are relevant and well-integrated into the text, though I suggest some minor clarifications to Figures 1 and 2 for improved readability. The article is self-contained and presents a coherent body of work. The introduction provides sufficient background, and relevant literature is cited to place the study in the broader context of urban ecology and avian foraging behavior. However, I recommend adding brief contextual information about the biogeography and introduction status of key species such as the Great-tailed Grackle and House Sparrow, especially given the grouping of other species into native and non-native categories. This addition would enhance consistency and depth in the interpretation of results.

Experimental design

The study addresses a relevant and timely research question, examining how impervious surface cover influences urban bird foraging behavior. The methods are generally well described, but several clarifications would strengthen the reproducibility of the study. Specifically, I suggest the authors provide additional details about:
• The type of food used at feeding sites (rationale behind using popcorn, especially due to the type of bird species that are more attracted due to dietary preferences),
• The timing of the trials (correction of time of day), as well as average temperature or weather conditions during the surveys, especially as Phoenix experiences extreme heat,
• The spatial limits around each survey site, including the sampling radius or area used for impervious surface estimation in each sampling point.
These methodological clarifications will improve transparency and allow for replication in future work.

Validity of the findings

The findings are generally valid and well presented, with sound statistical analyses. However, a critical limitation that should be addressed concerns the use of only the “first visitor” data. Many of the trials resulted in no visits, and this reduces the options (taking into account analytical power and ecological interpretability). I strongly suggest that the authors include the additional data they mention on secondary or subsequent visitors, even if they began collecting these after the first month. While statistical analysis might not be viable due to incomplete sampling, descriptive results—such as species identity, frequency, and co-occurrence patterns—can still provide valuable insights for behavioral ecology. The methodology section should simply clearly distinguish between the datasets used for inferential vs. descriptive analyses.
As mentioned before, the authors should address the potential consequences of using popcorn as the attractant in their limitations section. Certain species may be more attracted to this food type than others, potentially biasing visitor composition. This limitation is worth discussing in the interpretation of results and conclusions.

Additional comments

This study provides valuable insight into how urbanization gradients shape bird foraging behavior at human-derived food sources. It has potential for contributing to urban ecology and animal behavior literature. The experimental approach is effective, and the topic is increasingly relevant for urban wildlife providing ecosystem services and disservices. Clarifying methodological points and incorporating the additional data on subsequent visitors would significantly improve the study’s comprehensiveness and utility. I also appreciate the effort to distinguish species-level responses, and with a few enhancements to the species context and figure clarity, the manuscript will be much stronger. These are mostly minor or moderate revisions that do not detract from the quality of the work.

Annotated reviews are not available for download in order to protect the identity of reviewers who chose to remain anonymous.

---

## Round 0.2 · Minor Revisions

I am pleased to report that it is clear the authors did a great job revising the manuscript and addressing many of the reviewers' suggestions. I think this is shaping up to be a solid paper. Reviewer 1 has flagged a few more things to address, notably some issues with the explanation for the methods and the data visualisation for Figure 3. The reviewer has kindly offered some suggestion for an alternative graphing approach (see attached), some areas for a little more clarity on methods and reporting, how a few conclusions could be drawn, and a section that could be expanded to cover more avian taxa. I think these are all well-grounded suggestions, and I would encourage the authors to consider them. All in all, I think the work the authors have already done, and the addition of these current comments and suggestion, that this write up for this interesting study will have been markedly improved. Ya gotta love when the peer-review system works this well!

I suspect once this next round of edits and polishing is complete, we should have a shining paper to send out into the world. I am looking forward to seeing the next draft.

·

Basic reporting

The authors have done well to address my main concerns here.
The main concern I have with the revised version in terms of basic reporting is that Figure 3 is no-longer appropriate for the analyses conducted. Here is my suggestion for alternate presentation:
Because you used gaussian models the simple "lm" in geom_smooth should provide identical results to predict_results from ggeffects

ggplot()+
geom_point(data=dat, aes(x=impervious,
y = latency,
colour=species)) +
geom_smooth(method = "lm", data=dat, aes(x=impervious,y = latency))

I have included species as a factor because see section on validity of findings.

Experimental design

Updates to methods are well written and are mostly sufficient to replicate. However, in my attempts to replicate (as a follow through with the code provided above), I cannot get any of the Betas provided to match with those provided in the manuscript. Though they do not change the broad interpretation of the manuscript it would be helpful to provide clarity in the ms or provide supplemental code that explains:
- How N/A entries were handled in the analysis.
- Whether the logistic regression analyses were based on first birds at stations, or presence overall. And if the latter whether the data were subsetted to fit only the observations that other species' presence was noted. A further note on this is that though in the manuscript it states other species were noted after the first month of observations, but observations on the first day 7/7 recorded other visitors of house sparrows and house finches.

My other concern is that guild is not defined anywhere in the manuscript, and I've had to infer that the guilds are "small passerines (HOSP, HOFI), non-native doves, native-doves, grackles, and other natives". I usually think of guilds as species that obtain resources in similar ways, but I am not sure that is exactly the way they are being grouped here. Please elaborate.

Validity of the findings

See comments above regarding underlying data.

My main concern with these data were that the presence of small passerines + non-native doves in highly urban environments, plus the presence of non-native doves in less urban environments is driving the latency effect overall. I think it is worth noting in the main manuscript that the same general trend can be seen in the grackle data (though you cannot implement the full mixed effects model).

The conclusion section about certain birds being "messy" is overwraught given the focus of the manuscript was not on human perceptions but on bird behavioural mechanisms of community assembly.

Additional comments

The discussion of urban adapters in paragraph 258, centers only around doves, but the same effect was found with house sparrows and house finches exploiting urban environments. In the first part of paragraph 269 is also on the same topic and should be rolled together. The second part of that paragraph discussing methodological limitations ALSO hints at a mechanism of different community assembly.

I have provided further comments specifically on an marked up PDF.

Reviewer 2 ·

Basic reporting

The authors successfully improved the clarity of the sections that needed it, and the overall structure of the article is better as well, highlighting the relevance of their results. On the reviewed version of the manuscript, I wasn't able to see changes in the figures, but while my earlier suggestions stand (adding geographic coordinates to the map), it doesn't detract from the overall quality of the research.

Experimental design

No further comments as the authors expanded on their method section, clarifying previous caveats.

Validity of the findings

The new version of the Discussion section provides a robust framework for this and future research related to the subject, I don't consider any further changes are needed.

Additional comments

After reviewing the last version of the manuscript a single comment/question from the previous round remained (the one related to the surface area of the bounding boxes used on the i-Tree Canopy tool). Since I suspect different values were used for all sites, reporting the average (SD) would be enough. This suggestion is simply for comparison reference for future studies, thus if not considered, it doesn't change the overall clarity/quality of the rest of the manuscript.

---

## Round 0.3 · accepted · Accept

Well done. I think the manuscript is reading very well and presents the findings of this work clearly. They have address the reviewer comments, and either made the changes suggested, or provided justification for not. I think this manuscript has been appropriately improved by the peer-review process and I am happy to recommend it be accepted.

At this stage I typically conduct a final review for writing style, typos, and grammar, and then send the authors a Word .doc file with tracked changes for these minor tweaks. Upon conducting the final read through, I think this manuscript is pretty much in 'ship shape'. The only except being two things (that I suspect could be simply fixed during proofing - as they are not make or break. The first is on Line 53-54 in the sentence that reads "This advantage allows “urban exploiters” to outcompete “urban avoiders” in these environments." this should be contextualised and cited. Like explain what the terms mean and then cite the paper that first coined the terms (I think the citation is Blair, R. B. 1996. Land use and avian species diversity along an urban gradient. Ecological Applications, 6(2), 506-519). The other is you already say you have Ethics and provide the approval number in the Methods section and thus do not need to repeat this in the Acknowledgments (you could also thank your two reviewers in the Acknowledgments instead). Other than that I think this manuscript is all set.